# Accurate absolute free energies for ligand–protein binding based on non-equilibrium approaches

Vytautas Gapsys [1✉], Ahmet Yildirim [2], Matteo Aldeghi[3], Yuriy Khalak[1], David van der Spoel[4] & Bert L. de Groot [1✉]

The accurate calculation of the binding free energy for arbitrary ligand–protein pairs is a considerable challenge in computer-aided drug discovery. Recently, it has been demonstrated that current state-of-the-art molecular dynamics (MD) based methods are capable of making highly accurate predictions. Conventional MD-based approaches rely on the first principles of statistical mechanics and assume equilibrium sampling of the phase space. In the current work we demonstrate that accurate absolute binding free energies (ABFE) can also be obtained via theoretically rigorous non-equilibrium approaches. Our investigation of ligands binding to bromodomains and T4 lysozyme reveals that both equilibrium and non-equilibrium approaches converge to the same results. The non-equilibrium approach achieves the same level of accuracy and convergence as an equilibrium free energy perturbation (FEP) method enhanced by Hamiltonian replica exchange. We also compare uni- and bi-directional non-equilibrium approaches and demonstrate that considering the work distributions from both forward and reverse directions provides substantial accuracy gains. In summary, non-equilibrium ABFE calculations are shown to yield reliable and well-converged estimates of protein–ligand binding affinity.

[1] Computational Biomolecular Dynamics Group, Max Planck Institute for Biophysical Chemistry, Göttingen, Germany. [2] Department of Physics, Siirt University, Siirt, Turkey. [3] Vector Institute for Artificial Intelligence, Toronto, Canada. [4] Uppsala Centre for Computational Chemistry, Science for Life Laboratory, Department of Cell and Molecular Biology, Uppsala University, Uppsala, Sweden. ✉email: vgapsys@gwdg.de; bgroot@gwdg.de

The accurate prediction of binding free energies for protein–ligand complexes is of special interest in computational drug design and drug discovery campaigns more broadly. While there are various strategies to predict binding affinities, ranging from ligand-based chemoinformatics approaches[1] to structure-based docking[2], alchemical free energy calculations based on the first principles of statistical mechanics have recently been shown to achieve remarkable accuracy on a wide range of pharmaceutically relevant systems without the need for prior training data[3,4].

Common approaches rely on molecular dynamics (MD) simulations to explore the phase space of the ligand and target protein both in their bound and unbound states. A protein–ligand system represented at an atomistic level, together with the solvent and salt, may comprise more than a hundred thousand particles. Achieving sampling convergence for such large biomolecular systems presents a considerable challenge, especially if the full process of ligand binding to a target protein is to be simulated. However, if only the free energy difference between the end states—the bound and unbound states—is of interest, an alchemical approach can be employed to circumvent the computationally expensive physical path[5]. Indeed, alchemical methods exploit the control we have over the potential energy function to couple/decouple the ligand (or part of it) from the rest of the system, when in the solvent and in complex with a protein, and estimate the same free energy differences that otherwise would require explicit simulation of the physical binding process.

To facilitate the convergence of alchemical free energy estimates, relative (ΔΔG) rather than absolute (ΔG) Gibbs free energy differences are often sought. In this case, for a set of ligands of interest, the change in free energy difference (ΔΔG) with respect to one or a few reference ligands is estimated; the ΔG for these ligands is then recovered by adding the ΔΔG estimate to the known ΔG of the reference compounds. Because this setup requires the alchemical perturbation of only a small part of the ligand, in contrast to the coupling/decoupling of a whole molecule, it can in principle provide faster convergence. These calculations, when used in conjunction with state-of-the-art force fields, have been shown to provide predictions that, on average, deviate less than 1 kcal/mol from experimental measurements[3,4]. However, relative binding free energy (RBFE) calculations are not always well-suited to the problem at hand. For instance, when no known or suitable reference compound is available, when the aim is to identify the binding pose of a ligand, or when one is interested in ligand selectivity against different protein targets. In these situations, the direct calculation of the ligand–protein binding free energy is required.

Absolute binding free energy (ABFE) calculations require a different setup to that used for RBFE calculations[6,7]. The substantially larger perturbation of the system upon coupling/decoupling of the whole ligand needs longer sampling times to achieve convergence in comparison to the sampling needed to converge ΔΔG estimates. Although to our knowledge no large-scale studies on atomistic alchemical ΔG calculations have been published, a few small-scale studies hint at a lower accuracy compared to RBFE calculations. Although in the limit of infinite sampling both RBFE and ABFE calculations should return the same results, and thus provide the same accuracy, the empirical observation of lower accuracy for ABFE might be related to these higher sampling requirements. In the literature, ABFE calculation root mean squared error (RMSE) from the experimental measurements varied from 0.8–1.9 kcal/mol for T4 lysozyme inhibitors[8–10] to 2.3 kcal/mol for FKBP12 inhibitors[11]. Aldeghi et al. were able to reach a RMSE of 0.8 kcal/mol for 11 ligands binding to the first bromodomain of the bromodomain-containing protein 4 (BRD4(1))[7]. In a follow up study, the same authors investigated the selectivity of a compound (bromosporine) binding to a set of 22 bromodomains[12], and reported a RMSE of 1.9 kcal/mol.

The prevalent approach to carrying out alchemical free energy calculations is free energy perturbation (FEP) based on equilibrium simulations[13]. This method requires running a number of equilibrium simulations along the alchemical path at discrete steps. To control the system's position along the alchemical coordinate, a parameter λ is used to couple the Hamiltonians of the two physical end states. To then estimate the free energy difference between two states, the estimators derived by Zwanzig[13] or Bennet[14] (like Bennet's Acceptance Ratio, BAR) are typically used. The ΔG between the end states is then recovered by adding the free energy differences between all λ windows. A multistate version of BAR can also be used to obtain the ΔG value by considering all λ windows at once[15]. Another popular approach to estimate free energy differences from equilibrium alchemical simulations is thermodynamic integration (TI), where the average gradient of the Hamiltonian with respect to λ is integrated from λ = 0 to λ = 1[16].

A conceptually different approach relies on estimating ΔG from non-equilibrium transitions between the end states. In this case, equilibrium simulations are performed only for the two physical end states. Subsequently, rapid transitions through alchemical space are started, driving the system out of equilibrium and reaching the other physical state at the end of the alchemical morphing event. During such a transition, the force exerted along the alchemical λ coordinate is monitored and, afterward, integrated to obtain the work required for the transition. This method is the non-equilibrium, equivalent of the TI approach used for equilibrium calculations[17]. Having obtained a number of non-equilibrium work values, the free energy between the end states can be recovered by employing Jarzynski's equality[18] (for uni-directional transitions) or the Crooks Fluctuation Theorem[19] (for bi-directional transitions).

There is no clear consensus in the literature regarding the comparative efficiency of equilibrium and non-equilibrium approaches: probing various systems and using disparate efficiency measures, different authors have reached opposing conclusions[20–23]. Although the equilibrium methodology is more frequently employed, non-equilibrium methods have been used as well, for instance, in relative free energy calculations for amino acid mutations and protein thermostability[24,25], drug resistance[26–28], nucleotide mutations for protein–DNA interactions[29], as well as modifications of small organic molecules for assessing ligand–protein binding[4]. Recently, the applicability of non-equilibrium simulation to absolute binding free energy calculations has been explored for host-guest[30,31] and protein–ligand systems[32], and for the prediction of unbinding rate constants in protein–ligand complexes[33].

In this work, we assessed the applicability of non-equilibrium methods to ABFE calculations. To do this, we firstly selected two sets of protein–ligand complexes (Fig. 1) that have been previously studied with an equilibrium FEP approach enhanced by Hamiltonian replica exchange (HREX)[7,12]. The first dataset investigated the specificity of BRD4(1) against 11 inhibitors (Fig. 1a). The second dataset investigated the selectivity of bromosporine against 22 bromodomains (Fig. 1b). We compared the results from different methodologies: equilibrium FEP with and without HREX, as well as one- and bi-directional non-equilibrium TI. In addition, to explore the performance of alchemical approaches in a protein with a large conformational change between its apo and holo states, we calculated binding free energies for a set of 5 ligands interacting with the T4 lysozyme (L99A) protein. Overall, we found that non-equilibrium bi-directional approaches provide an equivalent performance to HREX-enhanced equilibrium FEP calculations.

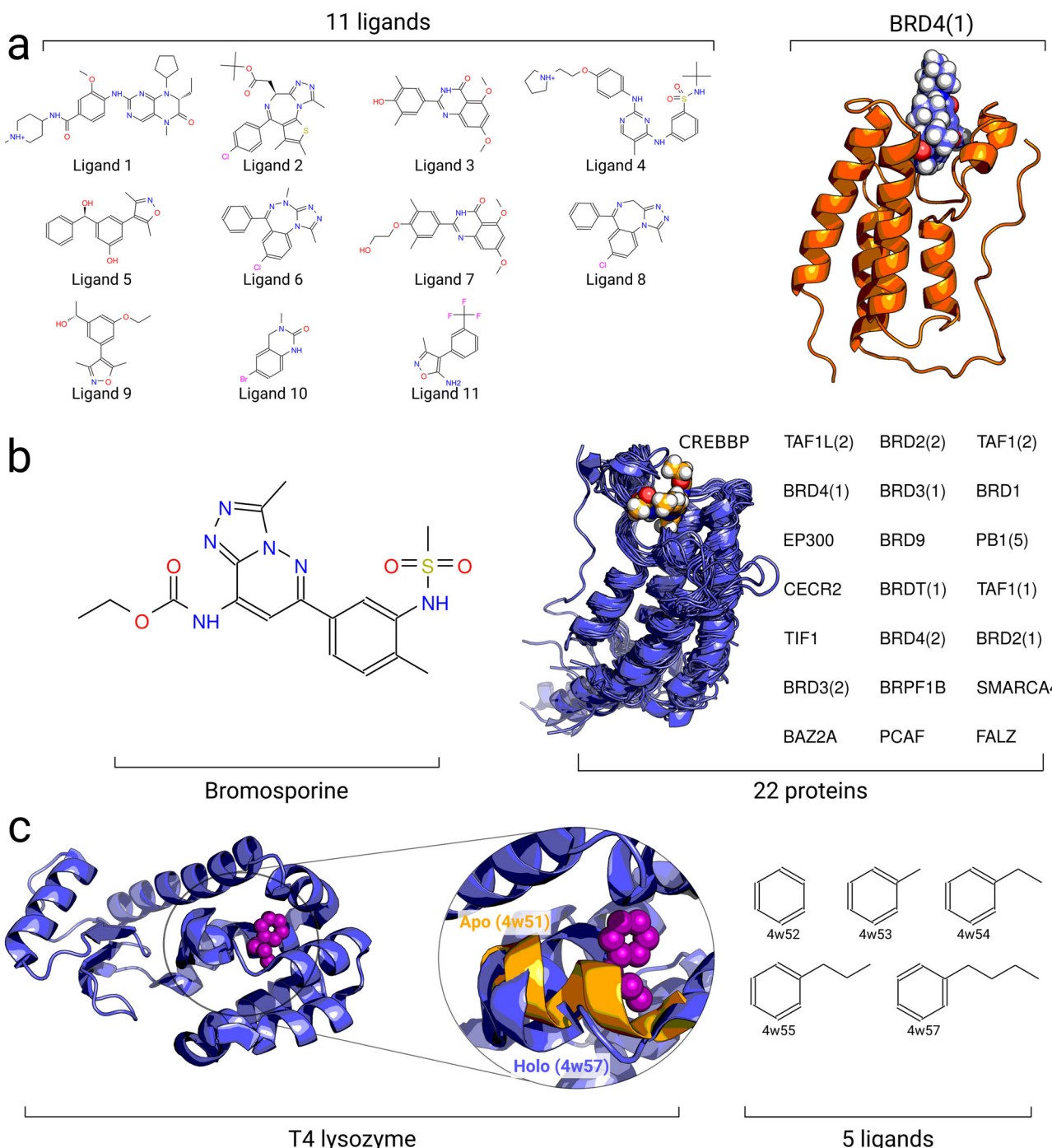

**Fig. 1 Overview of the investigated systems. a** Set of 11 ligands binding to the bromodomain BRD4(1) assembled in ref. [7]. **b** Set of 22 bromodomain complexes with the ligand bromosporine assembled in ref. [12]. **c** Crystallographic structure of T4 lysozyme complexed with a ligand (4w57). The inset shows an enlarged binding site of the aligned apo (4w51, orange) and holo (4w57, blue) structures, highlighting the major loop motion upon ligand binding. Also shown are 5 ligands binding to T4 lysozyme that were investigated in this work.

## Results

**Transition time tuning**. In this work, we studied how the equilibrium sampling time, as well as the number and length of non-equilibrium transitions, affect the convergence of the free energy estimates, while keeping the overall simulation time invested constant. Prior to starting this systematic analysis, we identified an optimal transition time for the non-equilibrium transitions, which would be held fixed for the rest of the investigation.

In a recent study, a search for an optimal non-equilibrium transition time in alchemical calculations of relative free energies identified a switching time of 80 ps to readily yield state-of-the-art accuracy[27]. It is, however, expected that the substantially larger perturbations required for the ABFE calculations might require longer transitions to reach convergence. To probe for an optimal transition time, we calculated the binding free energies of 7 ligands (a subset of those in a previous study[7]) to the BRD4(1)

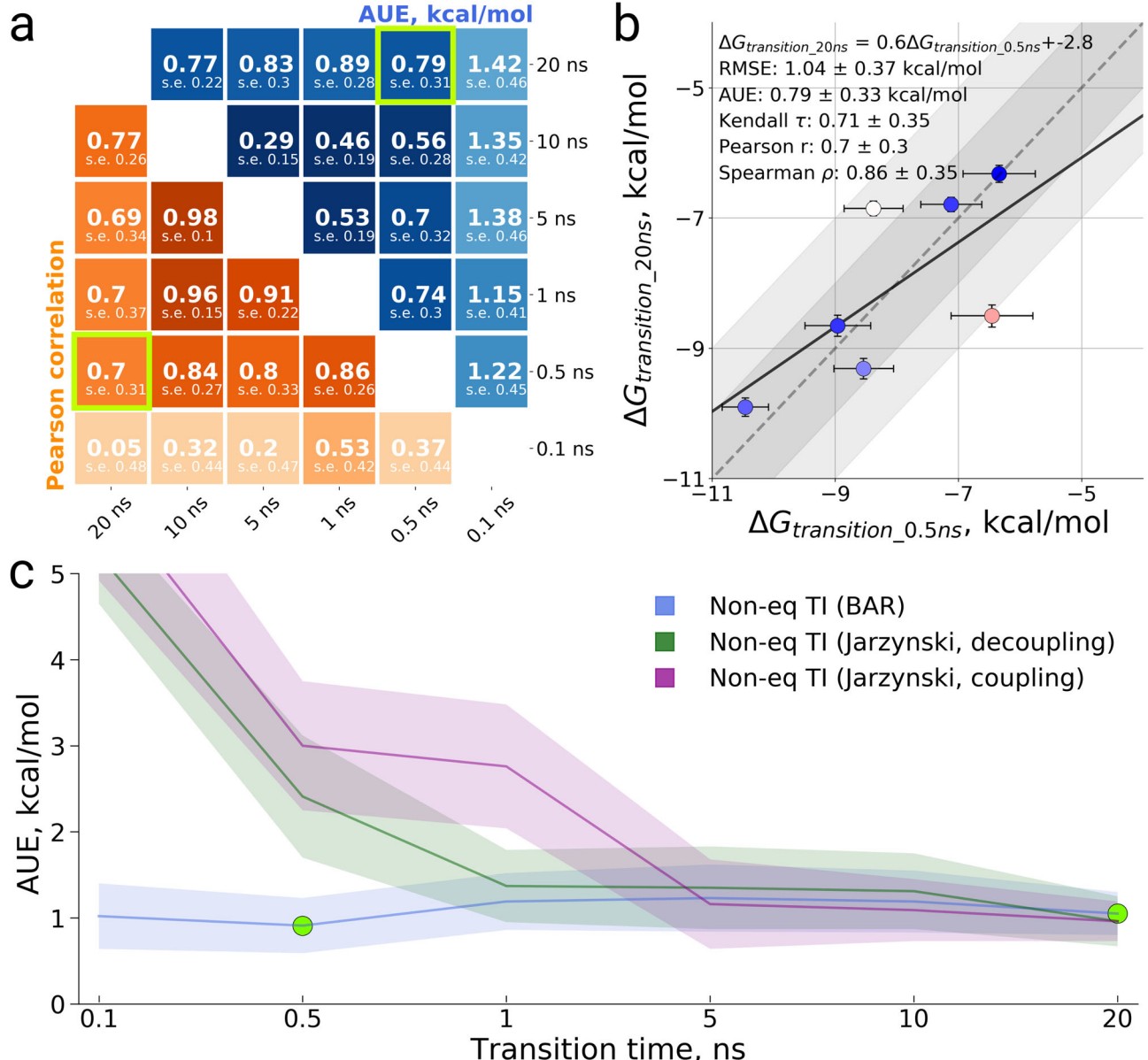

**Fig. 2 Binding free energy calculations for a subset of 7 ligands binding to BRD4(1). a** Comparison of the ΔG values estimated from simulations using different transition times in terms of average unsigned error (AUE) and Pearson correlation. The green frames in the plot mark the transition times for which the data in (**b**) is shown. **b** ΔG values from non-equilibrium TI calculations using 500 ps and 20 ns transitions plotted against each other. In the panel, we also provide a linear regression fit as well as RMSE, AUE, Kendall, Pearson, and Spearman correlations. **c** Average unsigned error (AUE) with respect to experimental measurements for the uni- and bi-directional non-equilibrium TI calculations. The bi-directional non-equilibrium TI protocol shows a consistent AUE with varying transition time, while uni-directional estimates based on Jarzynski's equality converge only with longer transition times. The green circles in the plot mark the transition times for which the data in (**b**) is shown. The uncertainties denote standard errors obtained by bootstrapping.

bromodomain using different lengths for the non-equilibrium transitions (Fig. 2, Fig. S1).

*Bias*. Firstly, we compared the ΔG values calculated with different non-equilibrium transition times to one another (Fig. 2a). The obtained results appear to converge to an average unsigned difference (AUE) below 1 kcal/mol for all transitions longer than 100 ps. Similarly, the Pearson correlation between any two transition times longer than 100 ps reaches the value of at least 0.69 (there is, however, a substantial standard error associated with this measure, as the investigated dataset is comprised of only 7 ligands). As expected, the shorter transitions yield estimates with larger associated uncertainties (Fig. 2b).

It is also useful to evaluate the convergence of the ΔG estimates for each transition time independently. In Fig. S2, we assess the convergence of each estimate using a convergence measure derived to quantify the work distribution overlap for bi-directional non-equilibrium free energy estimates[34]. This analysis differs from the one performed in Fig. 2a, as the convergence of each estimate is assessed independently, rather than by comparison to the calculations obtained using different transition times. The convergence measure used (a detailed description is provided in Supplementary Note 1) ranges from −1 to 1, with well-converged estimates returning a value close to 0. This analysis indicates that 100 ps transitions did not yield sufficient work distribution overlap to ensure a reliable free energy estimate (value

close to 1). A transition time of 500 ps, however, significantly improved convergence, with the exception of Ligands 2 and 6. In the results and analyses discussed in the next section, we will show that the lack of convergence for these two ligands can be alleviated by including more independent repeats while retaining 500 ps transition time (Fig. S3).

*Comparison to experiment.* Interestingly, the accuracy of the bi-directional free energy estimates averaged over the ligands, when compared to the experimental reference, is independent on the transition time (Fig. 2c). This, however, is in no contradiction to the earlier observation that transition time has some influence on the estimated ΔG: the free energy estimate of specific ligands can change depending on the length of the non-equilibrium transitions (Fig. S1). Fig. 2c rather indicates that, for the small chemical library at hand, the occasional gain or loss in accuracy due to longer switching times is likely to be within the level of statistical noise. While long sampling times of about 5 ns (Fig. 2c, Fig. S1) are needed to converge uni-directional free energy estimates based on Jarzynski's equality, the bi-directional estimates (non-eq TI BAR) converge with substantially shorter transitions of less than a nanosecond. These observations further support the notion that already sub-nanosecond transition times are sufficient to obtain ΔG estimates of low bias and high accuracy in reproducing experimental measurements. Longer transitions may still be required in specific instances (e.g., particularly large, flexible or charged ligands).

Considering the overall results of the transition time investigation, in the rest of this study, we used a transition time of 500 ps for all uncharged ligands. For charged ligands, where we observed convergence issues, we also explored the effect of longer transition times.

**BRD4(1) specificity**. To test the applicability of non-equilibrium free energy calculations to study ligand binding specificity, we calculated ΔG values for 11 ligands binding to the BRD4(1) bromodomain. This ligand set has been explored previously with HREX FEP[7] and represents a superset of that used in the non-equilibrium transition time analysis. Note that two ligands in this extended set had a net charge.

*Overall results and comparison to experiment.* From the overall method comparison (Fig. 3a, Table S1) the non-equilibrium TI approach with the 500 ps transitions time yielded a statistically indistinguishable accuracy compared to both FEP variants. Interestingly, HREX enhancement does not appear to give a significant boost to the FEP accuracy.

It is noteworthy that the accuracy of non-equilibrium TI calculations suffered from the estimate of one of the charged ligands: Ligand 4. The issue becomes more evident when monitoring the convergence of the simulations by quantifying the overlap of the work distributions (Fig. S3). Ligand 4 was clearly identified as lacking a well-converged estimate. For this ligand, we thus performed slower alchemical transitions. The transitions of 1 ns had the expected effect of enhancing the convergence (Fig. S3) and facilitating the agreement with the experimentally measured value (Fig. 3b). Extending the transition time to 2 ns, further increases the convergence, but the accuracy of the estimate changes only marginally.

Having observed that the improved convergence increased the calculated accuracy for Ligand 4, for the sake of consistency, we have also probed the effect of increasing the transition time for the other charged ligand (Ligand 1) too, which did not suffer from convergence issues with shorter (500 ps) switching time

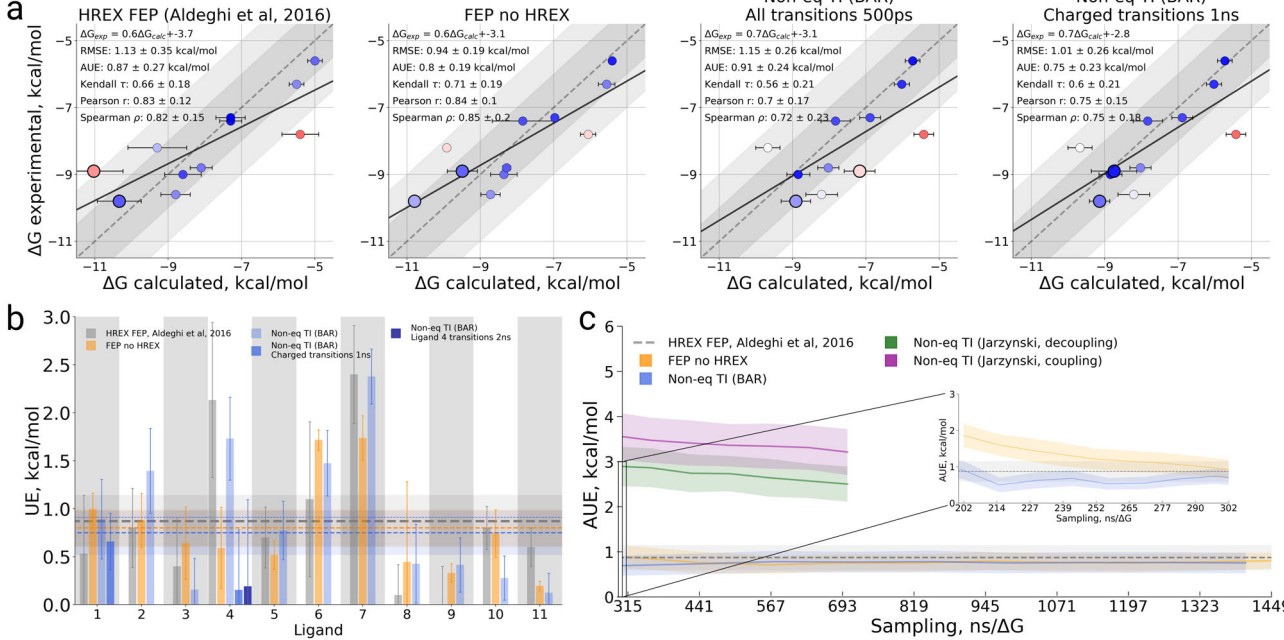

**Fig. 3 Binding free energy calculation summary for the BRD4(1) specificity study. a** Calculated vs experimental ΔG values for the HREX FEP[7], FEP without sampling enhancement, and non-equilibrium TI approaches. Two versions of the latter method are depicted: all transitions of 500 ps and extended 1 ns transitions for the charged ligands (Ligand 1 and Ligand 4). The larger circles in every panel denote Ligand 1 and Ligand 4. In the panels we also provide a linear regression fit as well as RMSE, AUE, Kendall, Pearson, and Spearman correlations. **b** Unsigned error (UE) by ligand for each of the considered protocols. **c** Average unsigned error (AUE) with respect to experimental measurement for varying sampling time invested in each of the approaches studied. The time reflects only the sampling invested in the protein–ligand coupling part of the calculation. The dashed line for HREX FEP[7] marks the accuracy of the estimate using the whole available sampling, i.e., it does not depict dependence on the sampling time. The inset shows the behavior of AUE for FEP and bi-directional non-equilibrium TI when less than 10% of the sampling production time is considered for the ΔG estimation. The uncertainties denote standard errors obtained by bootstrapping.

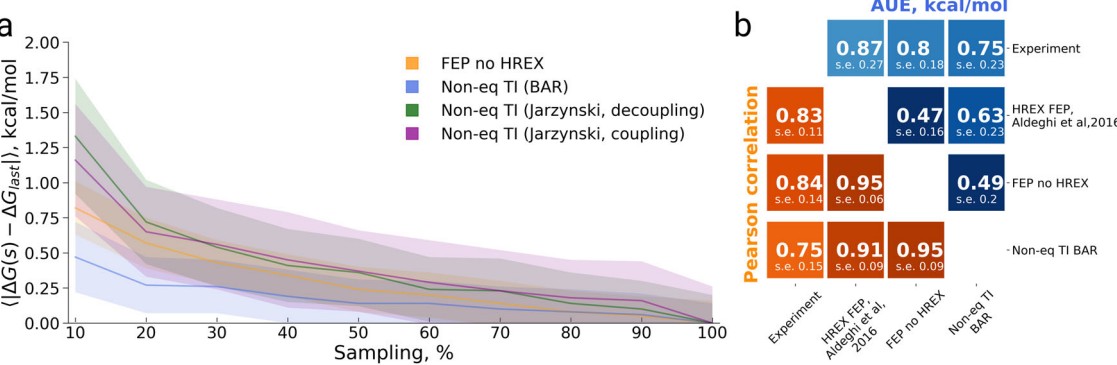

**Fig. 4 Bias analysis of absolute binding free energies for a set of 11 ligands binding to BRD4(1). a** Bias with respect to the final estimate using the whole available sampling against the invested sampling time. **b** Matrix comparing the calculation approaches, as well experimental measurement, in terms of Pearson correlation and average unsigned error (AUE). The uncertainties denote standard errors obtained by bootstrapping.

(Fig. S3). In this case, the longer transitions had only a minor, and statistically insignificant effect both on the convergence measure as well as on the accuracy of the $\Delta G$ estimate.

As described in the "Methods" section, all FEP and non-equilibrium TI calculations performed in this work employed the same amount of sampling as the HREX FEP approach used by Aldeghi et al.[7]. This allows for a convenient comparison of the convergence of each method studied here. Analysis of the unsigned error from experiment averaged over all 11 ligands shows rapid convergence of both the equilibrium FEP and non-equilibrium TI calculations (Fig. 3c). Using only 10% of the whole invested sampling time would already provide with an accuracy comparable to the final one that used 100% of the invested sampling time. The inset in Fig. 3c shows the accuracies achieved with less than 10% of sampling time, which highlights that the bi-directional non-equilibrium TI estimate converges significantly faster than FEP. The uni-directional estimates converged slower and, given equivalent sampling time, did not provide the accuracy of the other approaches.

*Bias.* While the AUE from experimental measurements provides insight into the convergence in terms of accuracy, it is also informative to inspect the bias of the free energy calculations. To assess this, we calculated the averaged unsigned difference of the estimated $\Delta G$ at a given time with respect to the final $\Delta G$ estimate (Fig. 4a). Monitoring $\Delta G$ estimates for each ligand separately provides further insight into the convergence trends (Fig. S4). The analysis of bias and ligand $\Delta G$ traces over time reveals a rapid convergence of the bi-directional non-equilibrium TI estimates. Using 10% of the overall sampling would result in an average bias of only 0.5 kcal/mol. The uni-directional estimators converge much more slowly. The equilibrium FEP approach also showed a larger bias than the bi-directional non-equilibrium TI across all sampling times.

A comparison of the $\Delta G$ estimates of the three computational protocols (HREX FEP, FEP, and non-equilibrium TI) reveals that they all yield similar results (Fig. 4b). Indeed, the approaches cluster tightly together when compared in terms of AUE and Pearson correlation. The results from computation and experiment are indistinguishable, as the differences between all the pair-wise comparisons fall within the uncertainty range.

**Bromosporine selectivity against multiple bromodomains.** The selectivity of bromosporine against 22 bromodomains has been explored previously in an ABFE calculation study that used HREX FEP[12]. The summary of the main results from that study is depicted in Fig. 5. Here, we have employed FEP without HREX,

as well as non-equilibrium TI for a comparative analysis on the same dataset. The total sampling time invested in each $\Delta G$ estimate was identical for all methods compared in Fig. 5.

*Overall results and comparison to experiment.* For this dataset, the HREX FEP outperformed FEP in absolute terms (RMSE and AUE), but had only a minor and statistically insignificant advantage over the non-equilibrium approach. In terms of correlation (Kendall, Pearson, and Spearman), the differences between FEP, HREX FEP, and non-equilibrium TI are not statistically significant.

A notable difference between the FEP methods and non-equilibrium TI comes from the estimated uncertainties for individual free energy estimates (Fig. 5a, b, Table S2). The error bars in case of the non-equilibrium TI are substantially larger than those for the FEP estimates. This effect is not, however, a feature of the particular free energy calculation method, but rather a manifestation of the stochastic nature of molecular dynamics simulations. For this selectivity dataset, non-equilibrium TI calculations were repeated multiple times (see "Methods") and the error bars were derived from independent simulation replicas. In contrast, FEP and HREX FEP used only a single simulation to estimate free energy, thus the associated error is that of the free energy estimator only. In fact, on the BRD4(1) specificity dataset, where repeated simulations were used for uncertainty estimates of all methods, uncertainties were comparable across all approaches (Fig. 3a, b).

In terms of convergence with respect to the experimental measurement (Fig. 5c), the methods follow a similar pattern as observed in the BRD4(1) specificity dataset. The accuracy of the bi-directional non-equilibrium TI approach does not change with an increase in sampling time. In fact, the estimate converges almost immediately after the equilibration (inset in Fig. 5c). FEP slowly converges to the same accuracy as HREX FEP and bi-directional non-equilibrium TI. The uni-directional non-equilibrium TI estimates converge significantly slower.

*Bias.* The bias for all the approaches is slightly larger for the bromosporine selectivity dataset (Fig. 6) when compared to the BRD4(1) specificity case (Fig. 4). The bi-directional non-equilibrium TI has the lowest bias: smaller than 1 kcal/mol even when using as little as 10% of the overall sampling data. The uni-directional estimates converge significantly slower and have a larger bias. The bias of the non-enhanced FEP method is slightly larger than that of the bi-directional non-equilibrium TI.

From the matrix comparing all methods and experiment in terms of AUE and Pearson correlation (Fig. 6b), the HREX FEP and non-equilibrium TI approaches appear to yield the most

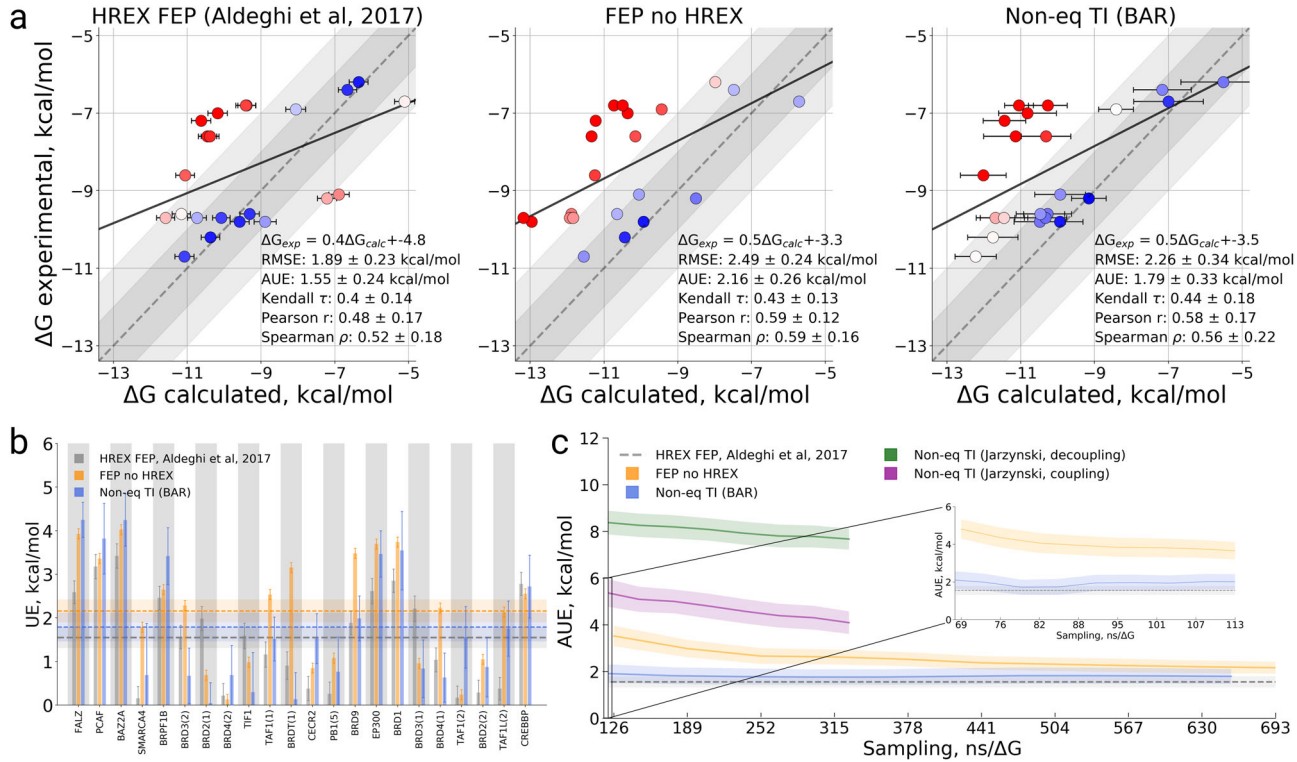

**Fig. 5 Binding free energy calculation summary for the bromosporine selectivity against 22 bromodomains. a** Calculated vs experimental ΔG values for the HREX FEP[12], FEP without sampling enhancement and non-equilibrium TI approach. In the panels, we also provide a linear regression fit as well as RMSE, AUE, Kendall, Pearson, and Spearman correlations. **b** Unsigned error (UE) by bromodomain for each of the considered protocols. **c** Average unsigned error (AUE) with respect to experimental measurement for varying sampling time invested for each of the investigated approaches. The time reflects only the sampling invested in the protein–ligand coupling part of the calculation. The dashed line for the HREX FEP value[12] marks only the accuracy of the estimate using the whole available sampling, i.e., it does not depict dependence on the sampling time. The inset shows the behavior of AUE for FEP and bi-directional non-equilibrium TI when less than 10% of the sampling production time is considered for the ΔG estimation. The uncertainties denote standard errors obtained by bootstrapping.

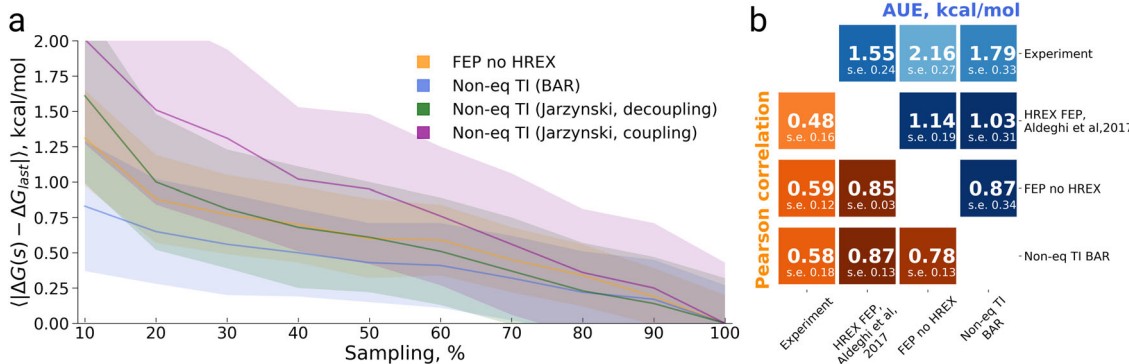

**Fig. 6 Convergence analysis of absolute binding free energies for bromosporine binding to 22 bromodomains. a** Bias with respect to the final estimate using the whole available sampling against the invested sampling time. **b** Matrix comparing the calculation approaches, as well experimental measurement, in terms of Pearson correlation and average unsigned error (AUE). The uncertainties denote standard errors obtained by bootstrapping.

similar ΔG estimates. Interestingly, these latter approaches show higher mutual agreement than the HREX FEP and non-enhanced FEP.

**Different Apo and Holo states: T4 Lysozyme (L99A).** The bromodomain investigations allow comparing the performance of alchemical approaches in computing absolute ligand–protein binding free energies. It is, however, important to probe whether the observed trends are transferable to other systems. In particular,

it is interesting to explore a protein exhibiting conformational differences between its apo and holo states.

A mutated L99A T4 lysozyme serves as a good test system for such a scenario: a large loop motion is observed at its binding site upon ligand binding (Fig. 1c). Lim et al.[35] demonstrated that insufficient sampling of this loop movement in MD simulations may result in inaccurate estimation of relative binding free energies. In their work, the authors used the FEP approach. In addition, they probed the effects of sampling enhancement by means of replica exchange solutetempering (REST)[36]. It was

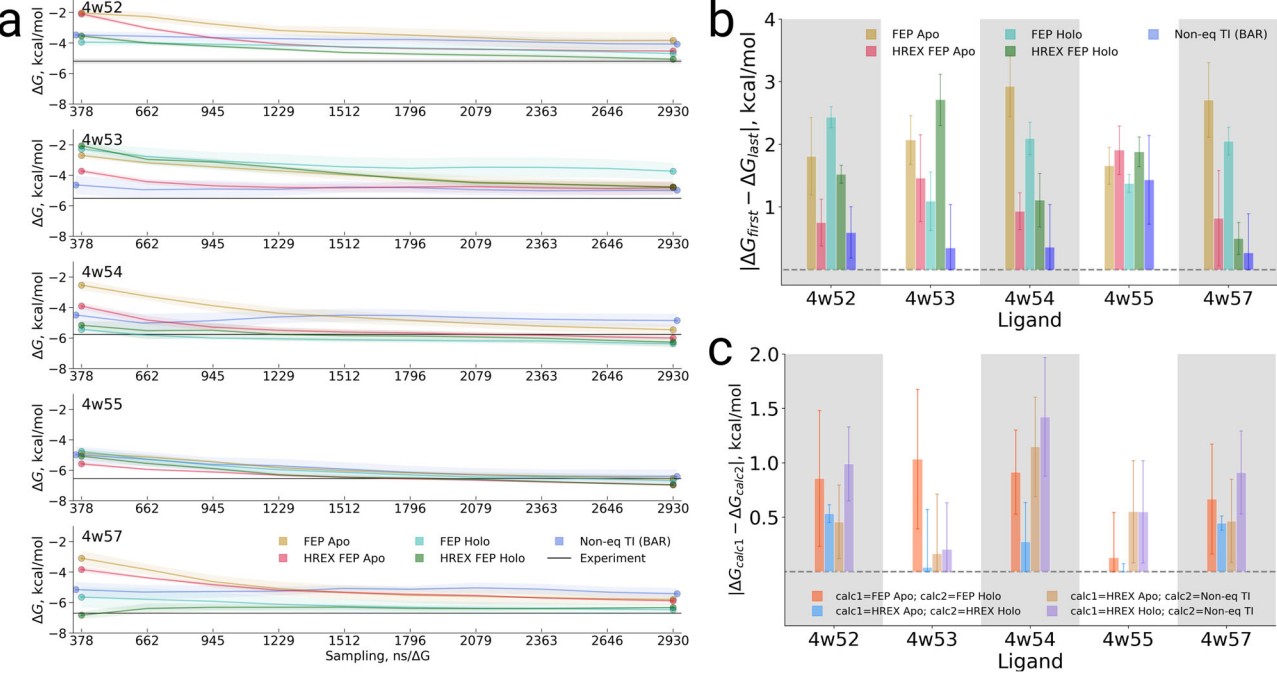

**Fig. 7 Free energy calculations of the ligands binding to T4 lysozyme (L99A). a** Free energy values calculated with different methods plotted against the sampling time. The time reflects only the sampling invested in the protein–ligand coupling part. **b** Absolute differences between the first free energy evaluation (cumulatively using approximately 380 ns for each method in the protein–ligand coupling part) and the last free energy evaluation (~2900 ns per method). **c** Absolute difference between the calculated free energy (final value, ~2900 ns per method) between several selected methods. The uncertainties denote standard errors obtained by bootstrapping.

observed that only long simulations for every discrete λ window (up to 55 ns) ensured equilibration of the binding site conformers. Given insufficient sampling, the simulations initiated with apo structures would yield binding free energies significantly different from those obtained when starting from holo structures.

In contrast to the FEP approach, the non-equilibrium protocol allows to easily initialize the apo and holo simulations with different starting structures (e.g., apo and holo conformers). This means that the ligand decoupling simulations can be initiated from the appropriate ensemble for the ligand bound to the protein (holo state), while the coupling simulations can be started from an ensemble initialized with an apo conformer.

We have calculated the binding free energies between T4 lysozyme and 5 ligands (Fig. 1c) for which experimentally measured affinities have been reported[37]. Visualizing ΔG changes with increased sampling time (Fig. 7a) illustrates that the FEP and HREX FEP estimates that are initialized from either apo or holo states need longer times to converge than the non-equilibrium TI initialized with both states. This effect is further summarized in Fig. 7b, where for each method and ligand an absolute difference between the first and last free energy estimate (with respect to the sampling time) is depicted. For example, the benzene (structure 4w52) FEP ΔG calculation, which was started from the apo conformer, changes by almost 2 kcal/mol from the first estimate (380 ns of cumulative sampling time) to the last estimate (2900 ns). Enhanced sampling HREX FEP calculation initiated with the same apo conformer facilitates the conver-gence: the first and last ΔG estimates differ by less than 1 kcal/mol. Interestingly, a faster convergence of HREX FEP is not a general feature, as HREX FEP started from the holo conformer shows a significantly slower convergence than FEP (also started from the holo conformer) for the structures 4w53 and 4w55. The non-equilibrium TI result changes the least with sampling time in comparison to other methods for all the considered ligands,

except for 4w55, where all methods show a comparable change in ΔG over time.

The simulations initialized from apo and holo structures do not necessarily converge to the same result in terms of the ΔG value (Fig. 7c, Table S3). The differences between the final free energy estimates from apo and holo simulations are significant in multiple investigated cases. FEP calculations without sampling enhancement converge to the same result only for the 4w55 case, while HREX facilitates the convergence between apo and holo simulations in most cases. The non-equilibrium TI method shows a diverse set of outcomes in this respect: for the 4w53 and 4w55 cases the result is similarly close to both HREX Apo and HREX Holo calculations while for the other ligands the result is closer to HREX Apo result.

**Discussion**
The presented study reveals that the non-equilibrium TI approach is a suitable method for protein–ligand absolute binding free energy calculations. The accuracy of this simulation protocol is on par with the equilibrium FEP even when the latter approach is enhanced by means of Hamiltonian replica exchange. In fact, sampling enhancement by means of HREX does not necessarily improve the accuracy of the estimated free energies, which is in agreement with a recent observation from the relative free energy calculation study[38].

For the explored systems, the overall accuracy is comparable to that observed in relative free energy calculations[3,4], where the gold standard is 1 kcal/mol in AUE. The observed high accuracy presents an interesting result, considering that the solvation free energies are highly sensitive to the inaccuracies in molecular mechanics force field parameterization[39]. Similar to relative free energy calculations, the accuracy varies depending on the system studied. In the current investigation we observed the AUE to be lower than 1 kcal/mol for the first dataset related to BRD4(1)

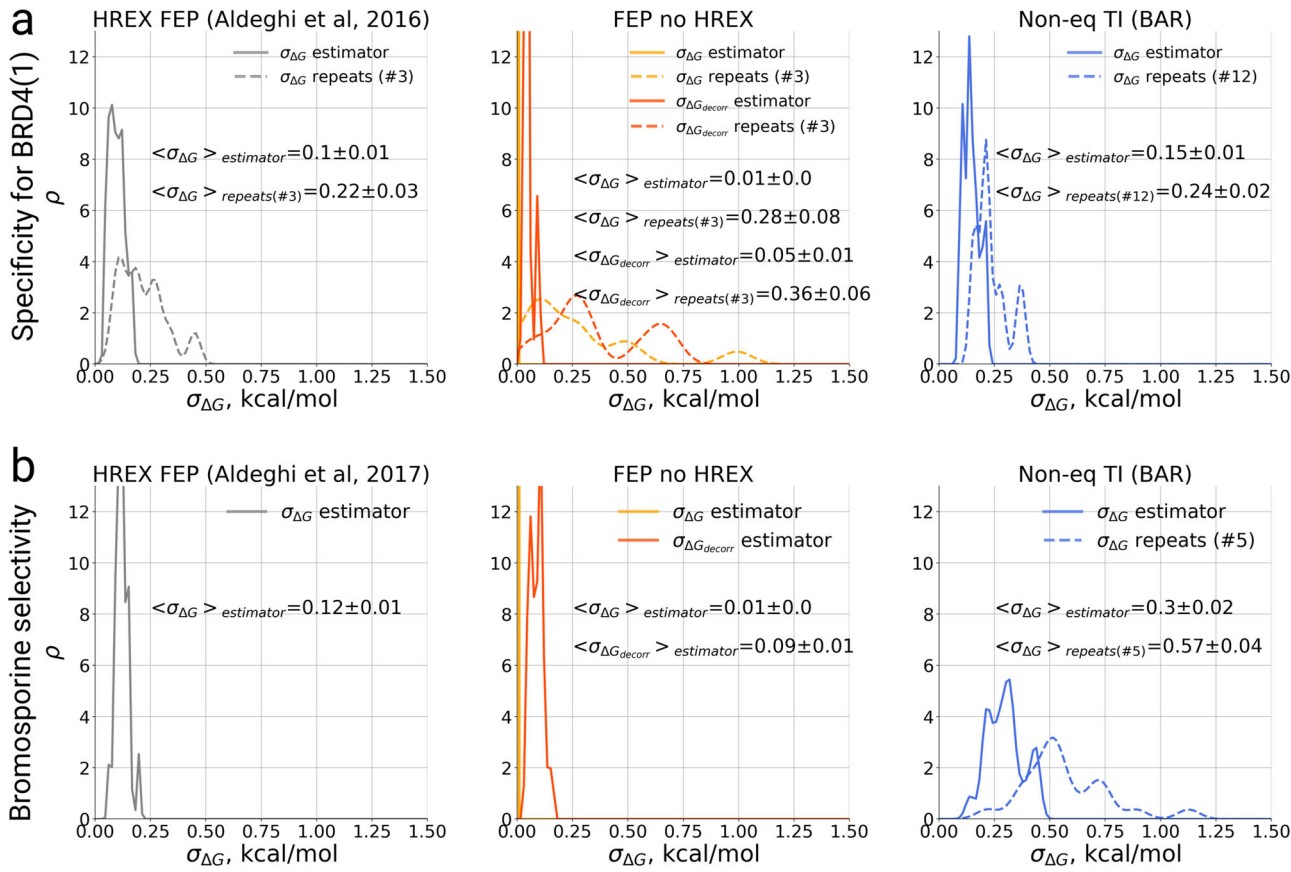

**Fig. 8 Distributions of uncertainty estimates for the ligand–protein coupling free energies.** The solid line depicts estimator uncertainty calculated from single simulation repeats. The dashed line shows the uncertainties obtained from independent simulation repeats. For the FEP no HREX case the uncertainty estimators from the whole data, as well as from uncorrelated samples ($\sigma_{\Delta G_{decorr}}$) are shown. **a** A set of 11 ligands binding to BRD4(1). **b** Bromosporine binding to 22 bromodomain proteins. The uncertainties denote standard errors of the mean.

specificity as well as for the T4 lysozyme ligand set, and an AUE slightly larger than 1.5 kcal/mol for the dataset related to bromosporine selectivity. In spite of the large perturbation involved in the (de)coupling of a whole ligand, the $\Delta G$ estimates obtained by the bi-direction non-equilibrium TI converged rapidly. Indeed, the overall accuracy reported could have been achieved with only ~10% of the whole invested sampling. The non-equilibrium approach also showed lower bias than the equilibrium FEP approach. This is in line with the recent observations for ABFE calculations in host-guest systems[40].

The uni-directional estimates showed significantly lower accuracy, slower convergence, and larger bias. Only with a substantially increased transition time the accuracy of the uni-directional estimates reached that of the bi-directional estimates. However, if possible, the sampling time that is spent in the alchemical transition to converge uni-directional estimates should preferably be invested in a more thorough exploration of the ensembles at the physical end states, which define the free energy difference.

For most of the protein–ligand systems explored in this work, the end-state sampling we invested appears to be sufficient to achieve converged and accurate binding free energy estimates. In this respect, bromodomains present a convenient test system: given the rigidity of their binding pockets, holo structures can be used to initialize simulations in the apo state without loss in $\Delta G$ prediction accuracy, even when short equilibrium simulations are used. The T4 lysozyme protein–ligand systems illustrate the opposite situation, where starting the simulations from apo or holo states significantly influences the calculation outcome. Here,

the end states (apo and holo) differ substantially, and more sampling time is required to obtain equilibrated structural ensembles.

Another important aspect to consider when comparing the calculation methods is the uncertainty of the estimated $\Delta G$ values. Fig. 8 summarizes the uncertainties for one branch of the thermodynamic cycle: ligand coupling to the protein ($\Delta G_{prot}$). We estimated standard errors using two different approaches. In the first, we calculated the standard error of multiple independent repeats: this approach quantifies the uncertainty in the estimates due to the stochastic sampling of different phase space regions in each simulation repeat. The standard error calculated this way will decrease by the factor $1/\sqrt{n}$ with the number of repeats $n$, thus in Fig. 8 we list explicitly the number of repeats considered for a corresponding uncertainty estimate. In the second approach, we estimated uncertainty from a single simulation repeat, using either the analytical uncertainty estimator for MBAR as implemented in *pymbar*[15] (for HREX FEP and FEP) or bootstrap (for the non-equilibrium TI). It is important to note that, in this case, the MBAR uncertainty estimator converges to one standard deviation of a normal $\Delta G$ distribution for a large number of samples, while the bootstrapped uncertainty estimators correspond to the standard errors of the mean of $\Delta G$. For the cases with multiple repeats, uncertainty propagation was applied. Overall, we observe that the uncertainty estimates obtained from single simulations underestimate the actual spread in the $\Delta G$ values one obtains when repeating the calculations. This effect is particularly pronounced for the non-enhanced FEP case, and slightly smaller for HREX FEP and non-equilibrium TI calculations (Fig. 8a). Because a single

calculation repeat was carried out for the bromosporine study with HREX FEP and FEP, we do not have uncertainty estimates from multiple repeats for these approaches (Fig. 8b). However, the uncertainty distributions for a single repeat closely resemble those observed for the BRD4(1) specificity study (Fig. 8a). The MBAR uncertainty estimator might be suffering from the presence of correlated data points in the analyzed sample. When only uncorrelated samples are used ($\sigma_{\Delta Gdecorr}$)[41] the estimated uncertainty increases significantly, albeit not to the level of that estimated from the independent repeats. The decorrelation procedure, however, does not have a significant effect on the uncertainty estimation from independent simulation repeats. All in all, these observations highlight the fact that one should not rely on uncertainty estimates from single simulation trajectories, but rather derive error bars from independently repeated calculations[42–44].

The apparent underestimation of the uncertainties further complicates the convergence analysis. Judging from the $\Delta G$ time traces (Figs. S4, S5), the bi-directional non-equilibrium TI estimates converge quickly and are stable over time. In comparison, the results obtained with equilibrium FEP converge slower. Even considering an apparently converged non-equilibrium TI result, a small (RMSE < 1 kcal/mol), yet statistically significant difference between the non-equilibrium TI and equilibrium HREX FEP calculation remains (Figs S6, S7). This disagreement might indicate that one of these approaches is still not sufficiently converged. On the other hand, considering that the uncertainty of the HREX FEP estimates is likely underestimated, the statistical significance of the difference between the estimates of the two methods might be an artefact caused by an underestimation of the uncertainty.

**Conclusions.** We have shown that ABFE calculations based on the non-equilibrium TI approach yield binding free energy estimates that are equivalent to those calculated with equilibrium FEP approaches, also when these are enhanced by means of Hamiltonian replica exchange. This observation held for all test sets explored here, which involved a total of 38 binding free energy estimates across 11 different small-molecules binding to BRD4(1) protein, bromosporine binding to 22 bromodomains, and 5 ligands binding to T4 lysozyme. These results thus demonstrate the feasibility of non-equilibrium MD simulations for the efficient estimation of ligand–protein binding affinities and lay the foundations to further explore these approaches as an alternative to equilibrium MD methods. To facilitate the setup of both equilibrium and non-equilibrium ABFE calculations, we provide a workflow as a part of the pmx package at https://github.com/deGrootLab/pmx. Overall, this study provides further evidence that alchemical calculations can yield accurate predictions of absolute protein–ligand binding free energies.

## Methods

**Simulated systems.** In this work we explored the accuracy of ABFE calculations in three scenarios. Firstly, we studied 11 ligands binding to BRD4(1) (Fig. 1a). The structures and topologies for the ligands and protein were identical to those used by Aldeghi et al.[7]. The experimental binding free energy values were also extracted from the same source. Of the 11 ligands, 9 were neutral and two carried a charge of +1 (ligands 1 and 4). Prior to calculating the free energy differences for the whole ligand set, a subset of these compounds was used in the search for an optimal alchemical transition time. This subset comprised ligands 2, 3, 5, 6, 7, 8, and 9 (Fig. 1a).

Secondly, we studied the ligand bromosporine complexed with 22 proteins (Fig. 1b). The structures, topologies, and experimental binding $\Delta G$ values were extracted from the previous investigation by Aldeghi et al.[12].

The third investigated system was T4 lysozyme (L99A) in complex with 5 ligands (Fig. 1c). The starting structures were based on the crystallographically resolved apo (pdb id 4w51) and holo (4w52, 4w53, 4w54, 4w55, and 4w57) structures[45].

**Free energy calculation.** We used a well-established procedure to calculate the absolute protein–ligand binding free energy. Following the description of the thermodynamic cycle outlined by Aldeghi et al.[7], the binding free energy calculation is split into a ligand solvation part ($\Delta G_{solv}$), a protein–ligand part ($\Delta G_{prot}$), and an analytically-computed contribution due to restraining the decoupled ligand to the protein ($\Delta G_{restr}$). The latter part applied the restraints described by Boresch et al.[6]. The protein and ligand atoms that were used for restraining, as well as the restraint strength, were identical to those used by Aldeghi et al.[7,12].

To be able to compare all the investigated approaches, we considered the $\Delta G$ as calculated without any additional post-hoc correction. While the long-range dispersion or finite size electrostatic corrections could modulate the overall predicted $\Delta G$ values, in the current work we concentrated only on the raw outcome of each of the investigated methods. Furthermore, we aimed at using an equivalent amount of sampling time for each computational strategy studied. For the equilibrium calculations, the same protocols as described by Aldeghi et al.[7,12] for HREX FEP were used. For the non-equilibrium calculations, the sampling time was distributed over the individual parts of the protocol to match the FEP simulation time as closely as possible.

**Equilibrium FEP.** For the FEP simulations of the first dataset (Fig. 1a, BRD4(1) specificity), 3 independent $\Delta G$ calculation repeats were performed. Ligand 11 was the only exception, in this case, two poses were considered separately, and two simulation repeats were performed for each pose (total of four $\Delta G$ calculations) in accord with the procedure used in ref. [7]. Each ligand solvation calculation was stratified into 31 windows, while the protein–ligand coupling path was divided into 42 windows. The stratification protocol followed that described in ref. [7]. Every window along the alchemical coordinate consisted of 0.5 ns equilibration in the NVT ensemble, followed by 1 ns in the NPT ensemble as equilibration, and finally 10 ns as production run. In total, 2.6 μs were invested in each $\Delta G$ estimate (1.7 μs for each pose of Ligand 11), matching that of the HREX FEP protocol in ref. [7].

The FEP protocol used in the simulations of the second dataset (Fig. 1b, bromosporine binding to 22 bromodomains) followed that used in ref. [12], matching the stratification strategy and sampling times. Five independent repeats (each using 31 windows) were performed to calculate $\Delta G_{solv}$, whereas a single repeat was performed to calculate $\Delta G_{prot}$ (using 42 windows). Each window consisted of an equilibration of 0.5 ns in the NVT ensemble, followed by 1 ns in the NPT ensemble, and a 15 ns production run, for a total of 3.3 μs per $\Delta G$ estimate, of which 0.7 μs were invested into the protein–ligand coupling simulations.

For the T4 lysozyme simulations, FEP and HREX FEP protocols were used by initializing the simulations with the apo (4w51) and respective holo structures[45]. The simulations used 16 windows to calculate $\Delta G_{solv}$ and 21 windows for $\Delta G_{prot}$. Simulations for both legs of the thermodynamic cycle were started with a position restrained 0.5 ns NVT equilibration followed by a 1 ns NPT equilibration. The production runs were performed for 15 ns per window in case of the solvation free energy calculations and for 45 ns per window for the protein–ligand complex simulations. Each $\Delta G_{solv}$ and $\Delta G_{prot}$ calculation was repeated three times. The exchange frequency between replicas for the HREX FEP followed closely the protocol by Aldeghi et al.:[7] swaps were attempted every 1000 steps by performing $3 \times 10^6$ exchanges at each interval.

**Non-equilibrium TI.** In the non-equilibrium TI calculations, we tried to closely match the overall sampling time invested in the FEP protocols per $\Delta G$ estimate. Each solvation and protein–ligand coupling free energy calculation consisted of an equilibrium and non-equilibrium simulation parts. For the BRD4(1) specificity dataset, equilibrium simulations consisted of a 0.5 ns simulation in the NVT ensemble, followed by a 10 ns production simulation in the NPT ensemble for both end states: ligand coupled and decoupled from the system. Subsequently, 96 snapshots were extracted from each equilibrium trajectory equidistantly (the first 0.4 ns from the trajectories were discarded for equilibration), from which 0.5 ns non-equilibrium transitions were started and performed in both coupling and decoupling directions. This procedure was repeated 9 times for the ligand solvation part and 12 times for the protein–ligand coupling part. An exception is Ligand 11, where 6 solvation and 8 protein–ligand coupling runs were performed for each of the two poses to ensure equivalent sampling time to that invested in the FEP approaches.

For the bromosporine selectivity dataset, we used longer equilibrium simulations of 15 ns (the first 5 ns were discarded for equilibration) and 100 non-equilibrium alchemical transitions of 0.5 ns each. Solvation free energy calculations were repeated 20 times, while protein–ligand calculations were repeated 5 times for each protein–bromosporine complex.

T4 lysozyme calculations used six independent repeats for the solvation and for protein–ligand simulations. Ligand in water equilibrium runs were of 15 ns, while protein–ligand runs were of 45 ns. In each case 5 ns were discarded for equilibration. 100 and 400 non-equilibrium transitions of 0.5 ns each were performed to calculate $\Delta G_{solv}$ and $\Delta G_{prot}$, respectively. For the protein–ligand simulations, the ligand coupling runs were initialized with an apo structure of the protein (pdb id 4w51), while the decoupling runs were started from respective holo structure.

For all the studied cases, when calculating bias and $\Delta G$ deviation from experiment over time, the $\Delta G$ estimates from the non-equilibrium TI were

obtained by truncating the equilibrium transitions and using a fraction of transitions.

Before performing the calculations described above, we studied the effect of different non-equilibrium transition times for a subset of 7 ligands, part of the first BRD4(1) specificity dataset. These calculations used equilibrium simulations of 20 ns, from which 50 snapshots were extracted after discarding the first 50 ns for equilibration. The lengths of the alchemical transitions studied were of 0.1, 0.5, 1, 5, and 10 ns for the calculation of $\Delta G_{solv}$. For the calculation of $\Delta G_{prot}$, also 20 ns transitions were tested. The binding $\Delta G$ estimate was calculated by considering $\Delta G_{solv}$ and $\Delta G_{prot}$ derived from non-equilibrium transition of equivalent length, with the only exception of the longest (20 ns) transitions used to calculate $\Delta G_{prot}$, which were combined with the results for $\Delta G_{solv}$ that used 10 ns transitions.

The overall simulated time invested in this study (for both equilibrium FEP and non-equilibrium TI calculations) was of 149 µs. This amounts to ~36 CPU years on an Intel Xeon 3.5 GHz CPU with one NVIDIA Quadro P600 GPU. In this study, we aimed to keep the overall simulation time comparable between all approaches investigated, i.e., the same compute time was invested for each of the methods. The main differences in applying FEP, HREX FEP, and non-equilibrium TI methods comes from the type of jobs needed for efficient simulations. For the FEP calculations, simulation at each discrete window along the alchemical coordinate can be performed on a separate node, thus all the jobs can be submitted to a cluster without the need to consider any inter-connections between them. In contrast, HREX FEP requires communication between replicas, which necessitates running all HREX jobs at once on a large, single node or multiple nodes with fast connections. The non-equilibrium TI method consists of two steps, which are performed sequentially. First, equilibrium simulations at the physical end-states are performed: this step, in terms of the compute resource selection, is similar to FEP. Second, many short (e.g., 0.5 ns) non-equilibrium simulations are run. Here, the non-equilibrium approach allows for more freedom in the choice of compute nodes, allowing to achieve high throughput performance. In fact, these short simulations can be run sequentially on a single fast node or be spread across the whole cluster in an embarrassingly parallel manner.

**MD simulation parameters**. The simulation parameters used were identical to those used in the earlier bromodomain studies, which are publicly available[7,12]. The proteins were parameterized with the Amber99SB-ILDN force field[46,47]. The General Amber force field (GAFF)[48] was used to parameterize the ligands. The systems were solvated with TIP3P[49] water, consistently with the simulation setup in refs. [7,12]. $Na^+$ and $Cl^-$ ions were added to neutralize the system. In the bromosporine and T4 lysozyme simulations (Fig. 1b, c), ions were added to reach a salt concentration of 150 mM, while for BRD4(1) no additional salt was added for consistency with the setup used by Aldeghi et al.[7]. Production simulations were performed at constant temperature (298.15 K) and pressure (1 bar), using Langevin dynamics and the Parrinello–Rahman barostat[50], respectively. The stochastic dynamics integrator with a friction constant of $1 ps^{-1}$ and a 2 fs integration time step was used. Bonds involving hydrogen atoms were constrained by means of the LINCS algorithm[51], with a bond restraining order of 6. The Particle Mesh Ewald algorithm[52,53] was used to treat long-range electrostatics with a spline order of 6, a relative tolerance of $10^{-6}$, Fourier grid spacing of 0.1 nm, and a direct space cutoff of 1.0 nm for the first dataset (Fig. 1a) and 1.2 nm for the second and third datasets (Fig. 1b, c). For the simulations of the first dataset, the van der Waals interactions were switched off between 0.9 and 1.0 nm, while a shifted van der Waals potential was used for simulations of the second and third datasets with an interaction cutoff of 1.2 nm. In the equilibrium FEP simulations, only van der Waals interactions were soft-cored as proposed by Beutler et al.[54]. In the non-equilibrium TI calculations, both the electrostatic and van der Waals interactions were soft-cored as described by Gapsys et al.[55] for the bromodomain simulations and by Beutler et al.[54] for T4 lysozyme. All the equilibrium simulations were performed with Gromacs[56] 2018. The non-equilibrium transitions with the modified soft-core function were performed with Gromacs 4.6 (in these simulations we set the direct space electrostatic cutoff to 1.2 nm for both datasets, for compatibility with the neighbor list cutoff used in ref. [7]).

For the set of simulations performed to study the effect of non-equilibrium transition time, a slightly modified set of parameters was used. Namely, the equations of motion were integrated with a smaller time step of 1 fs. The LINCS bond restraining order was set to 12. A direct space electrostatic interaction cutoff of 1.2 nm was used with a Fourier grid spacing of 0.12 nm. The van der Waals interactions were switched off between 1.0 and 1.1 nm.

**Free energy and uncertainty estimation**. For the equilibrium FEP calculations, free energy estimates and associated uncertainties were calculated using the multistate Bennet's acceptance ratio (MBAR)[15] estimator as implemented in *pymbar* and the Alchemical Analysis tool[57]. For analysis, the first 10% of the simulations was discarded as equilibration. For the MBAR estimates, we decided not to use a sample decorrelation procedure[41]. This choice was made after noticing that performing the time series analysis occasionally results in discarding a large portion of the overall data leaving only a few points for analysis. This, in turn, may result in estimates deviating significantly from the results of other methods, as well as from the experimental measurement. Bypassing the time series analysis resolved the issue of such outliers, yet the uncertainties of the individual estimates appear to be under-estimated. We, therefore, also provide a more detailed analysis on the uncertainty estimation.

For the non-equilibrium TI calculations, bi-directional free energy estimates were calculated using a maximum likelihood estimator[58] that is equivalent to the Bennet's acceptance ratio (BAR) as derived for the ensembles sampled at equilibrium[14]. Uni-directional free energy estimates were calculated using Jarzynski's equality[59]. These analyses were carried out using the pmx[60] tool. Uncertainties were obtained via bootstrap.

Where multiple independent calculation repeats were performed, the standard error for a $\Delta G$ estimate was calculated by incorporating both the estimator uncertainty as well as the variance of the repeated calculations. For each repeat, the distribution of $\Delta G$ values was assumed to be normal, with mean equal to $\Delta G$ estimate and standard deviation equal to the bootstrapped uncertainty. The final uncertainty estimate considering all repeats was calculated as a standard error across all the normal distributions.

The agreement between the calculations and experiment was quantified by the average unsigned error (AUE), the root mean squared error (RMSE), and the Pearson, Kendall, and Spearman correlations. The error bars for these measures were obtained by bootstrap, by taking into account both the variation in the dataset, as well as the uncertainties associated with the individual $\Delta G$ estimates. This was achieved by a combination of parametric and non-parametric bootstrap in which we both resampled $\Delta G$ values as well as their associated uncertainty. At each bootstrap iteration, samples from the dataset were selected at random with replacement, where for each sample a $\Delta G$ value was drawn from a normal distribution according to its estimated mean and standard error. Throughout the work we report uncertainties as: estimator $\pm$ standard error. In some cases this representation violates the bounds of the estimator, as e.g. a correlation larger than 1. We note that this is merely a consequence of the uncertainty representation, none of the actual values violate such bounds.

To calculate $\Delta G$ estimates using a fraction of the production sampling time ($\Delta G$ plots against sampling time), for the non-equilibrium TI calculations we considered shorter equilibrium simulations, with the number of alchemical transitions being reduced accordingly, while retaining a transition time of 0.5 ns. In case of FEP (and HREX FEP for T4 lysozyme simulations), simulation time was truncated accordingly to evaluate $\Delta G$ over time. When depicting the $\Delta G$ (or its AUE) against time (e.g., in Figs. 3, 5, 7), the *x*-axis denotes the total sampling time, of which the first fraction is used for equilibration as described for each protocol individually. This means that the shortest simulation time considered in each of these figures is not zero as it represents the total amount of simulated time used for equilibration before the production runs.

## Data availability
The free energy estimates with uncertainties reported in Figs. 2–8 are provided as supporting material (Supplementary Data 1) together with this publication.

## Code availability
The simulations were performed with Gromacs (LGPL-2.1 license): https://gitlab.com/gromacs. Analysis of the non-equilibrium free energy calculation data was performed with pmx (LGPL-3.0 license): https://github.com/deGrootLab/pmx. Analysis of the equilibrium free energy calculation data was performed with Alchemical Analysis (MIT License): https://github.com/MobleyLab/alchemical-analysis.

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

## Acknowledgements

The Swedish research council is acknowledged for financial support to D.v.d.S. (grant 2013-5947) and for a grant of computer time (SNIC2017-12-41) through the High Performance Computing Center North in Umeå, Sweden, and the PDC Center for High Performance Computing at the Royal Institute of Technology, Stockholm, Sweden. V.G. and B.L.d.G. were supported by the BioExcel CoE (http://www.bioexcel.eu), a project

funded by the European Union (Contract H2020-INFRAEDI-02-2018-823830). M.A. was supported by a Postdoctoral Research Fellowship of the Alexander von Humboldt Foundation. Y.K. was supported by the Vlaams Agentschap Innoveren & Ondernemen (VLAIO) project number HBC.2018.2295, "Dynamics for Molecular Design (DynaMoDe)".

## Author contributions

V.G. and A.Y. performed the calculations and analyzed the data. V.G., A.Y., M.A., D.v.d.S., and B.d.G. designed the study and interpreted the results. V.G., A.Y., M.A., Y.K., D.v.d.S., and B.d.G. contributed to writing and reviewing the manuscript.

## Funding

## Competing interests

The authors declare no competing interests.
