## [Peer Review File · Communications Chemistry]

Reviewers' comments:

Reviewer #1 (Remarks to the Author):

The authors outline the application of non-equilibrium free energy methods to absolute binding affinity predictions, which is demonstrated on two systems.

Their results are well presented against previous results, which clearly demonstrates the strengths of this method on par to currently more widely used algorithms within the field. They demonstrate accuracy comparable to those expected for relative free energy calculations, which at ~ 1 kcal/mol makes the method a useful tool for drug discovery.

The statistical analysis of the results is robust, and allows for fair comparison between methods, and the methods employed are clearly explained in the text.

Scientific points

"Fig S2. Convergence measure for the bi-directional (BAR) non-equilibrium TI based free energy estimates" it is not clear to me what the convergence measure is, or how it's calculated. The caption indicates 1 is bad and 0 is good, which is helpful, but I don't know how to interpret the negative value.

Choice of switching time - the choice of 500ps switching time (1ns for charged compounds) is justified through agreement with experiment, however it would be useful to see the robustness of the choice in switching times using other analytical methods, such as looking at the overlap of the forwards and -reverse work distributions would be useful. There may not always be experimental values to benchmark with, and a discussion of other checks that are possible would be helpful in knowing if the protocol and method is behaving appropriately.

Minor points

charge of+1 (needs a space)

Aldeghi et al.7. (remove fullstop)

"Ligand 1 and Ligand4" in Figure 3 caption (needs a space)

This is a well written publication, with solid and convincing results, presented clearly. This publication will be of high interest to the wider field.

Reviewer #2 (Remarks to the Author):

Gapsys et al. report a validation study of non-equilibrium free-energy calculations to determine protein-ligand binding free energies. One dataset for affinity, of dissimilar ligands binding to the same bromodomain receptor, and for selectivity, one ligand binding to multiple receptors, is investigated.

In general the topic of the study is of current interest as free energy calculation methods have experienced a boom in practical applications over the last years. The goal of the authors, to show that (some) non-equilibrium free energy methods perform as well as more commonly used equilibrium approaches is an interesting addition to our experience with free energy methods and well supported by the data presented. I'd argue that the findings, while novel, are not particularly surprising. In the limit of sufficient sampling, any properly set up free energy calculation should converge to the same result if the underlying potential energy function is the same, independent of method. However, as the authors outline some optimized parameters on how to conduct such calculations in practice, the manuscript may allow for wider use of non-equilibrium calculations in the future, which would represent a useful addition to the computational chemistry toolkit. I would therefore recommend publication with minor modifications addressing the following comments:

My main concern is a deeper discussion of the system selected. All results presented are for bromodomain proteins. These are fairly rigid receptors for which binding site conformational changes play a relatively small role. This makes me wonder how generalizable the results are to other pharmaceutically relevant targets. One could suspect, even though I have no proof either way, that a more flexible binding pocket poses bigger problems for non-equilibrium than equilibrium free energy methods since the system has less time to adapt to an alchemical transformation. Expanding the study to additional proteins would be a significant effort, but this issue should at least be discussed in the manuscript.

In a similar fashion, one thing I found surprising regarding the comparison to experimental values is the lack of a large offset for calculated absolute binding free energies. Even a well converged simulation is likely orders of magnitudes too short for a full relaxation to the apo protein. This should result in a more or less uniform offset in the results which seems not to be present here. Is there a good explanation of this apart from bromodomains possibly having relative low such reorganisation energies?

More minor comments:

Figure 1B shows bromosporin bound to only one of the 22 proteins selected (which one?). An overlay of the various receptor structures could be interesting to judge their similarity.

In the methods section, for non-equilibrium TI (but not for the equilibrium one) a 0.5 ns NVT equilibration phase is apparently followed immediately by a longer NPT production run. How was the density equilibration monitored? Earlier production snapshots may not be at a converged system density.

Why was the number of simulation repeats reduced for the ligand with multiple poses considered? As long as the two poses don't interconvert during the simulation, they should be treated as independent ligand binding events and sampled the same way as any other compound.

The authors use TIP3P water, a decades old water model that was parameterized I believe even before the advent of modern long-range electrostatics calculations. While it remains a common choice in MD simulations, many newer water models are available. Where any of them considered?

Why was a physiological salt concentration only used in the selectivity calculation dataset? I would imagine the mix of charged and neutral compounds in the affinity set to also benefit from a realistic

ionic strength?

The legend of Fig 2B gives two dG values, 0.6 and +2.8, which I first didn't understand and only on second reading realized represent a linear fit of the data points. This should be labeled in the caption.

In figure 3C and elsewhere, the authors show results that would have been obtained with fewer (e.g. 10% of) sampling. How was the amount of data reduced, by running fewer non-equilibrium simulations or running them shorter? In general, even at 10% of scaling, simulations still amount to dozens or hundreds of nanoseconds of simulation. Since the data is already there, would it be possible to show how further reduction to say 2% or 5% affects the results, especially since the two best performing methods still show fairly low errors at the minimum amount of sampling analyzed? Seeing when the result quality really breaks down would be interesting.

This might be nitpicking, but "In terms of correlation (...), FEP and non-equilibrium TI outperformed HREX FEP, however, here too the differences are not statistically significant.". I'd argue that if the difference is not significant, then one method is not outperforming the other.

Reviewer #3 (Remarks to the Author):

The study by Gapsys et al. deals with the calculation of binding free energies for ligand-protein pairs using non-equilibrium approaches. Computer aided drug discovery is a hot topic and calculating drug binding affinities is of crucial importance.

The study compares the match of the experimental binding free energies by calculations based on conventional equilibrium sampling methods with those calculations based on non-equilibrium approaches. The experimental values and the protocol for the equilibrium calculations as well as the choice of the test examples of 11 ligands and 22 proteins are taken from Aldeghi et al. (Chemical Science 2016).

The study is carried out carefully, the drawn conclusions are well supported by the data and critical points are discussed. The manuscript is well organized and written. The results are presented in a clear way well supported by figures graphs and Tables supporting the key conclusion that calculations based on the non-equilibrium TI approach yield binding free energy estimates that are equivalent to those calculated with equilibrium FEP approaches, also when these are enhanced by means of Hamiltonian replica exchange.

It is clear that the results are comparable but the authors should discuss more clearly the advantages why one should be interest to use a non-equilibrium approach if the results are as good as with equilibrium approaches. Like the needed computer time and the usability could be compared. For example in the methods section it is only stated how much computer time was used over all (~25

CPU years on an Intel Xeon 3.5 GHz CPU with one NVIDIA Quadro P600 GPU) this should be split up for the different approaches.

As stated at the beginning of the Introduction, the key motivation for the work is the accurate prediction of binding free energies for protein–ligand complexes. To refer to this motivation a short discussion how good the calculated free binding energies match the experimental values overall should be added to the conclusion.

Overall the study is of high relevance to the chemical computation community laying the foundation for future investigations of binding free energies through non-equilibrium approaches.

In this light, the authors should cite the recent work on protein-ligand interactions also using non-equilibrium MD, namely Wolf et al. "Estimation of Protein–Ligand Unbinding Kinetics Using Non-Equilibrium Targeted Molecular Dynamics Simulations." *Journal of chemical information and modeling* 59.12 (2019): 5135-5147.

Minor note:

Figure 1 and Table S1 shows 11 ligands whereas in the conclusion 12 ligands are mentioned.

Reviewer 1

Scientific points

"Fig S2. Convergence measure for the bi-directional (BAR) non-equilibrium TI based free energy estimates": it is not clear to me what the convergence measure is, or how it's calculated. The caption indicates 1 is bad and 0 is good, which is helpful, but I don't know how to interpret the negative value.

The convergence estimator is based on the work by Hahn & Then, 2010, PRE (10.1103/PhysRevE.81.041117). It was observed that the overlap between the forward and reverse work distributions (U) can be defined in two different ways based on the Fermi functions for the forward ($b_{\Delta F}$) and reverse ($t_{\Delta F}$) process. The overlap can be defined using the first moments $\hat{U} = \overline{t_{\Delta F}} = \overline{b_{\Delta F}}$, as well as the second moments $\hat{U}'' = \frac{n_0}{N} \overline{t_{\Delta F}^2} + \frac{n_1}{N} \overline{b_{\Delta F}^2}$. Based on these overlap estimates a convergence measure is defined as $a = \frac{\hat{U} - \hat{U}''}{\hat{U}}$. The measure is defined over the range of $-1 < a \leq 1$. Since the overlap estimate based on the second moments converges slower than \hat{U} , for the unconverged ΔF estimates, a has a value close to 1. With the increased number of observations, the overlap increases, thus facilitating the convergence and bringing the a value closer to 0. Upon reaching convergence, a fluctuates around 0: thus the negative values with small absolute value indicate of an already converged estimate.

We have added a more detailed description of the convergence measure to the Supporting Information (page 3) and adjusted the main text.

We used a convergence measure specifically derived for bi-directional non-equilibrium free energy estimates, which ranges from -1 to 1 and where well-converged estimates should return a value close to 0 (a detailed description of the convergence measure is provided in the Supporting Information).

Choice of switching time - the choice of 500ps switching time (1ns for charged compounds) is justified through agreement with experiment, however it would be useful to see the robustness of the choice in switching times using other analytical methods, such as looking at the overlap of the forwards and -reverse work distributions would be useful. There may not always be experimental values to benchmark with, and a discussion of other checks that are possible would be helpful in knowing if the protocol and method is behaving appropriately.

That is a very good point, as in prospective studies the choice of the sampling time cannot depend on the experimental values. In the current work we have also primarily based the decision to use 500 ps switching time on the comparison of the ΔG values obtained with varying transition times, i.e. without relying on the experimental reference (section **Transition time tuning: Bias**). Here, we ensured that the calculated results do not differ substantially with the further increased transition time (Figure 2A). Now, in addition to the aforementioned check, we have evaluated the convergence for different switching times based on the overlap of the distributions.

A description has been added to the maintext and a new Figure S2 has been added to the Supporting Information.

It is also useful to evaluate the convergence of the ΔG estimates for each transition time independently. In the Figure S2 we assess the convergence of each estimate using a convergence measure derived to quantify the work distribution overlap for bi-directional non-equilibrium free energy estimates. This analysis differs from the one performed in Figure 2A, as the convergence of each estimate is assessed independently, rather than by comparison to the calculations obtained using different transition times. The convergence measure used (a detailed description is provided in the Supporting Information) ranges from -1 to 1, with well-converged estimates returning a value close to 0. This analysis indicates that 100 ps transitions did not yield sufficient work distribution overlap to ensure a reliable free energy estimate (value close to 1). A transition time of 500 ps, however, significantly improved convergence, with the exception of Ligands 2 and 6. In the results and analyses discussed in the next section, we will show that the lack of convergence for these two ligands can be alleviated by including more independent repeats while retaining 500 ps transition time (Figure S3).

Minor points

charge of+1 (needs a space)

Fixed.

Aldeghi et al.7. (remove fullstop)

Fixed.

"Ligand 1 and Ligand4" in Figure 3 caption (needs a space)

Fixed.

Reviewer 2

My main concern is a deeper discussion of the system selected. All results presented are for bromodomain proteins. These are fairly rigid receptors for which binding site conformational changes play a relatively small role. This makes me wonder how generalizable the results are to other pharmaceutically relevant targets. One could suspect, even though I have no proof either way, that a more flexible binding pocket poses bigger problems for non-equilibrium than equilibrium free energy methods since the system has less time to adapt to an alchemical transformation. Expanding the study to additional proteins would be a significant effort, but this issue should at least be discussed in the manuscript.

That is a valid and interesting point. The binding site flexibility is expected to influence the efficiency of the probed methods, however, it is not necessarily true that non-equilibrium approaches would suffer most from undersampling. For the cases where the apo and holo protein states are significantly different, all the methods would face a challenge of taking into account the binding site reorganisation. In fact, the non-equilibrium approach can utilize a convenient feature of initializing the equilibrium simulations from apo and holo states independently for evaluation of the same free energy difference, while equilibrium methods require to make a choice: apo or holo starting structure.

To closer explore these questions we have now added a set of calculations of T4 lysozyme complexed with five ligands. This protein is known to undergo a substantial loop motion at its active site upon ligand binding.

A section **Different Apo and Holo states: T4 Lysozyme (L99A)** is now added to the main text.

In a similar fashion, one thing I found surprising regarding the comparison to experimental values is the lack of a large offset for calculated absolute binding free energies. Even a well converged simulation is likely orders of magnitudes too short for a full relaxation to the apo protein. This should result in a more or less uniform offset in the results which seems not to be present here. Is there a good explanation of this apart from bromodomains possibly having relative low such reorganisation energies?

It is a correct note by the reviewer: the bromodomains have been shown to behave well with respect to the absolute free energy estimation [refs 7, 8], which indicates that the simulations explore sufficient phase space volume for both, apo and holo states. This is well explained by the high similarity between the experimentally resolved apo and holo bromodomain structures. For example, high resolution x-ray structures for the bromodomain apo (pdb id 2oss) and holo (3mxf) states differ only by ~ 0.2 Å. In the reply to the previous reviewer's question, we have described the now added investigation of a ligand set binding to T4 lysozyme. For these cases, the apo and holo states differ substantially, thus a larger range of offsets between the calculated and experimental ΔG values is observed.

We now address this point in more detail in the Discussion section.

For most of the protein-ligand systems explored in this work, the end-state sampling we invested appears to be sufficient to achieve converged and accurate binding free energy estimates. In this respect, bromodomains present a convenient test system: given the rigidity of the binding pocket, holo structure of the protein can be used to initialize simulations in the apo state without loss in ΔG prediction accuracy, even when short equilibrium simulations are used. The T4 lysozyme protein-ligand systems illustrate the opposite situation, where starting the simulations from apo or holo states significantly influences the calculation outcome. Here, the end states (apo and holo) differ substantially, and more sampling time is required to obtain equilibrated structural ensembles.

More minor comments

Figure 1B shows bromosporin bound to only one of the 22 proteins selected (which one?). An overlay of the various receptor structures could be interesting to judge their similarity.

Figure 1B was now updated to show 22 aligned protein structures.

In the methods section, for non-equilibrium TI (but not for the equilibrium one) a 0.5 ns NVT equilibration phase is apparently followed immediately by a longer NPT production run. How was the density equilibration monitored? Earlier production snapshots may not be at a converged system density.

To avoid the issue of not fully converged system density during the NPT production runs, we have discarded the initial part from the simulated trajectory. This information is now included in the manuscript.

Subsequently, 96 snapshots were extracted from each equilibrium trajectory equidistantly (the first 0.4 ns from the trajectories were discarded for equilibration)

For the bromosporine selectivity dataset we used longer equilibrium simulations of 15 ns (the first 5 ns were discarded for equilibration)

Why was the number of simulation repeats reduced for the ligand with multiple poses considered? As long as the two poses don't interconvert during the simulation, they should be treated as independent ligand binding events and sampled the same way as any other compound.

The number of repeats was adjusted to invest an equivalent sampling time for this ligand to that used by Aldeghi et al, 2016, thus ensuring fair comparison of the results from different methods. The ΔG value for this ligand was calculated by taking into account two poses and the cumulative number of repeated calculations in this case was even larger than for any other ligand. We now include a more detail description of this in the paper.

Ligand 11 was the only exception, in this case, where two poses were considered separately, and two simulation repeats were performed for each pose (total of four G calculations) in accord with the procedure used in [7].

An exception is Ligand 11, where 6 solvation and 8 protein-ligand coupling runs were performed for each of the two poses to ensure equivalent sampling time to that invested in the FEP approaches.

The authors use TIP3P water, a decades old water model that was parameterized I believe even before the advent of modern long-range electrostatics calculations. While it remains a common choice in MD simulations, many newer water models are available. Where any of them considered?

It is true that many newer water models have been developed improving on some of the deficiencies of TIP3P. The goal of the current work, however, was to evaluate computational approaches in computing absolute binding free energies. For a consistent comparison with the HREX FEP method used by Aldeghi et al [refs 7 and 12], the simulation setup was identical to the earlier simulations, thus TIP3P model was retained. We have now updated the methods section explaining the choice of TIP3P water model.

The systems were solvated with TIP3P water consistently with the simulation setup in 7, 12.

Why was a physiological salt concentration only used in the selectivity calculation dataset? I would imagine the mix of charged and neutral compounds in the affinity set to also benefit from a realistic ionic strength?

That is a valid point and using a physiological salt concentration would certainly be the preferred setup option. As already mentioned, in the current work, we aimed to compare the performance of several methods, therefore, we tried to retain the simulation setup as similar to the previous calculations [refs 7 and 12] as possible. This is now also emphasized in the text.

In the bromosporine simulations (Fig. 1B), ions were added to reach a salt concentration of 150 mM, while for BRD4(1) no additional salt was added for consistency with the setup used by Aldeghi et al. [7]

The legend of Fig 2B gives two dG values, 0.6 and +2.8, which I first didn't understand and only on second reading realized represent a linear fit of the data points. This should be labeled in the caption.

An explanation was added to the figure captions: Fig 2, Fig 3 and Fig 5.

In the panel we also provide a linear regression fit as well as RMSE, AUE, Kendall, Pearson and Spearman correlations.

In figure 3C and elsewhere, the authors show results that would have been obtained with fewer (e.g. 10% of) sampling. How was the amount of data reduced, by running fewer non-equilibrium simulations or running them shorter? In general, even at 10% of scaling, simulations still amount to dozens or hundreds of nanoseconds of simulation. Since the data is already there, would it be possible to show how further reduction to say 2% or 5% affects the results, especially since the two best performing methods still show fairly low errors at the minimum amount of sampling analyzed? Seeing when the result quality really breaks down would be interesting.

The amount of data was reduced by using fewer repeats, but retaining a consistent alchemical transition time of 0.5 ns. It is true that even 10% of the overall simulation time may reach a hundred of nanoseconds. This is due to the specific setup of the current work: here we aimed to set the total invested simulation time to be equivalent among the probed methods. In calculating the total sampling time we included also the equilibration steps that amount to a non-negligible quantity, e.g. to equilibrate 42 discrete FEP windows we used 0.5 ns NVT and 1 ns NPT runs, which already sums up to 63 ns. This equilibration time, thus dictates the lower time limit up to which we can extract ΔG estimates. In the manuscript we have now added these additional explanations on the analysis. In Figures 3 and 5, we are now also showing a more fine grained analysis of the first 10% of the production sampling time.

To calculate ΔG estimates using a fraction of the production sampling time (ΔG plots against sampling time), for the non-equilibrium TI calculations we considered shorter equilibrium simulations, with the number of alchemical transitions being reduced accordingly, while retaining a transition time of 0.5 ns. In case of FEP (and HREX FEP for T4 lysozyme simulations), simulation time was truncated accordingly to evaluate ΔG over time. When depicting the ΔG (or its AUE) against time (e.g., in Figures 3, 5, 7), the x-axis denotes the total sampling time, of which the first fraction is used for equilibration as described for each protocol individually. This means that the shortest simulation time considered in each of these figures is not zero as it represents the total amount of simulated time used for equilibration before the production runs.

Using only 10% of the whole invested sampling time would already provide with an accuracy comparable to the final one that used 100% of the invested sampling time. The inset in Fig 3C highlights that the bi-directional non-equilibrium TI estimate converges significantly faster than FEP.

The accuracy of the bi-directional non-equilibrium TI approach does not change with an increase in sampling time, in fact, the estimate converges almost immediately after the equilibration (inset in Fig 5C).

This might be nitpicking, but "In terms of correlation (...), FEP and non-equilibrium TI outperformed HREX FEP, however, here too the differences are not statistically significant.". I'd argue that if the difference is not significant, then one method is not outperforming the other.

Rephrased.

In terms of correlation (Kendall, Pearson, and Spearman), the differences between FEP, HREX FEP and non-equilibrium TI are not statistically significant.

Reviewer 3

It is clear that the results are comparable but the authors should discuss more clearly the advantages why one should be interested to use a non-equilibrium approach if the results are as good as with equilibrium approaches. Like the needed computer time and the usability could be compared. For example in the methods section it is only stated how much computer time was used over all (25 CPU years on an Intel Xeon 3.5 GHz CPU with one NVIDIA Quadro P600 GPU) this should be split up for the different approaches.

This is a good suggestion. In the current study we aimed to keep the overall simulation time comparable between the investigated approaches, i.e. the same compute time was invested for each of the methods. The main differences in applying FEP, HREX FEP and non-equilibrium TI methods comes from the type of jobs that need to be generated. For example, HREX FEP requires communication between the replicas and this necessitates running all HREX jobs at once on a large single node or multiple nodes with a fast connection. Non-equilibrium TI allows for much more freedom in the choice of compute nodes for a high throughput: most of the jobs only need to run a short (e.g. 0.5 ns) independent simulations, thus depending on the available cluster architecture these small jobs can be run sequentially on a single fast node or be spread in an embarrassingly parallel manner.

A discussion on this is now included in the manuscript.

In this study, we aimed to keep the overall simulation time comparable between all approaches investigated, i.e. the same compute time was invested for each of the methods. The main differences in applying FEP, HREX FEP and non-equilibrium TI methods comes from the type of jobs needed for efficient simulations. For the FEP calculations, simulation at each discrete window along the alchemical coordinate can be performed on a separate node, thus all the jobs can be submitted to a cluster without the need to consider any inter-connections between them. In contrast, HREX FEP requires communication between replicas, which necessitates running all HREX jobs at once on a large, single node or multiple nodes with fast connections. The non-equilibrium TI method consists of two steps, which are performed sequentially. First, equilibrium simulations at the physical end-states are performed: this step, in terms of the compute resource selection, is similar to FEP. Second, many short (e.g. 0.5 ns) non-equilibrium simulations are run. Here, the non-equilibrium approach allows for more freedom in the choice of compute nodes, allowing to achieve high throughput performance. In fact, these short simulations can be run sequentially on a single fast node or be spread across the whole cluster in an embarrassingly parallel manner.

As stated at the beginning of the Introduction, the key motivation for the work is the accurate prediction of binding free energies for protein–ligand complexes. To refer to this motivation a short discussion how good the calculated free binding energies match the experimental values overall should be added to the conclusion.

That is a good point. We have now extended the discussion and conclusions part.

*For the explored systems, the overall accuracy is comparable to that observed in relative free energy calculations where the gold standard in the AUE is 1 kcal/mol. Similar to relative free energy calculations, the accuracy varies depending on the system studied. In the current investigation we observed the AUE to be lower than 1 kcal/mol for the first dataset related to BRD4(1) specificity *as well as for the T4 lysozyme ligand set*, and an AUE slightly larger than 1.5 kcal/mol for the second dataset related to bromosporine selectivity.*

*This observation held for *all* test sets explored here, which involved a total of 38 binding free energy estimates across 11 different small-molecules *binding to BRD4(1) protein, bromosporine binding to 22 bromodomains and 5 ligands binding to T4 lysozyme*. These results thus demonstrate the feasibility of non-equilibrium MD simulations for the efficient estimation of ligand-protein binding affinities and lay the foundations to further explore these approaches as an alternative to equilibrium MD methods. *The alchemical calculations are demonstrated to be capable of yielding accurate predictions of absolute protein-ligand binding free energies.**

Overall the study is of high relevance to the chemical computation community laying the foundation for future investigations of binding free energies through non-equilibrium approaches. In this light, the authors should cite the recent work on protein-ligand interactions also using non-equilibrium MD, namely

Wolf et al. "Estimation of Protein–Ligand Unbinding Kinetics Using Non-Equilibrium Targeted Molecular Dynamics Simulations." Journal of chemical information and modeling 59.12 (2019): 5135-5147.

We thank the reviewer for bringing this work to our attention. It is indeed a relevant reference and we have included it in the paper.

. Recently, the applicability of non-equilibrium simulation to absolute binding free energy calculations has been explored for host-guest[30,31] and protein-ligand systems[32], and for the prediction of unbinding rate constants in protein-ligand complexes[33].

Minor note:

Figure 1 and Table S1 shows 11 ligands whereas in the conclusion 12 ligands are mentioned.

In the conclusions we summarize all the explored systems combining all the calculated free energies for the proteins-ligands depicted in Fig 1A (11 ligands) and Fig 1B (1 ligand), as well as the newly added T4 lysozyme system in Fig 1C (5 ligands). We agree that the original wording was confusing and we have now rephrased the conclusions section.

REVIEWERS' COMMENTS:

Reviewer #2 (Remarks to the Author):

The authors have addressed all questions raised in review and the addition of the T4 lysozyme system offers another interesting example. I recommend publication of the manuscript.

Reviewer #3 (Remarks to the Author):

The authors have addressed all those points we raised in the first step and addressed them satisfactorily.